

# Resolving ambiguity in natural language for enhancement of aspect-based sentiment analysis of hotel reviews

Asma Nadeem[1], Malik Muhammad Saad Missen[1], Mana Saleh Al Reshan[2,3], Muhammad Ali Memon[4], Yousef Asiri[5], Muhammad Ali Nizamani[4], Mohammad Alsulami[3,5] and Asadullah Shaikh[2,3]

[1] Information Technology, Islamia University, Bahawalpur, Punjab, Pakistan
[2] Information System, Najran University, Najran Province, Saudi Arabia
[3] Emerging Technologies Research Lab (ETRL), Najran University, Najran Province, Saudi Arabia
[4] Information Technology, University of Sindh, Jamshoro, Sindh, Pakistan
[5] Computer Science, Najran University, Najran Province, Saudi Arabia

Corresponding author
Asadullah Shaikh,
shaikhasad@hotmail.com

## ABSTRACT

In the ever-expanding digital landscape, the abundance of user-generated content on consumer platforms such as Booking and TripAdvisor offers a rich source of information for both travellers and hoteliers. Sentiment analysis, a fundamental research task of natural language processing (NLP) is used for mining sentiments and opinions within this vast reservoir of text reviews. A more specific type of sentiment analysis, *i.e.*, aspect-based sentiment analysis (ABSA), is used when processing customer reviews is required. In ABSA, we aim to capture aspect-level sentiments and intricate relationships between various aspects within reviews. This article proposes a novel approach to ABSA by introducing a novel technique of word sense disambiguation (WSD) and integrating it with the Transformer architecture bidirectional encoder representations from Transformers (BERT) and graph convolutional networks (GCNs). The proposed approach resolves the intriguing ambiguities of the words and represents the review data as a complex graph structure, facilitating the modeling of intricate relationships between different aspects. The combination of bidirectional long short-term memory (BiLSTM) and GCN proves effective in capturing inter-dependencies among various aspects, providing a nuanced understanding of customer sentiments. The experiments are conducted on the RABSA dataset (an enhanced and richer hotel review data collection), and results demonstrate that our approach outperforms previous baselines, showcasing the effectiveness of integrating WSD in ABSA. Furthermore, an ablation study confirms the significant contribution of the WSD module to the overall performance. Moreover, we explore different similarity measures and find that cosine similarity yields the best results when identifying the real sense of a word in a given sentence using WordNet. The findings of our work and future work related to our work create lots of interest for people in the tourism and hospitality industry. This research gives another boost to the concept of the potential of NLP techniques in sentiment analysis. It emphasizes that if we combine the potential of NLP techniques along with state-of-the-art machine learning frameworks, we can shape the future of this field.

## INTRODUCTION

In this era which is dominated by online platforms, consumers are more habitual of voicing their opinions and share experiences, significantly affecting the reputations of businesses, especially in the competitive hospitality industry.

The impressive growth of platforms for booking hotels and sharing reviews (like www. booking.com and www.tripadvisor.com) has resulted in an exponential increase in user-generated content. Understanding opinions and preferences expressed in this user-generated content is a crucial aspect of natural language processing (NLP) (*Oralbekova et al., 2023*). The task of sentiment analysis (SA) (*Mukherjee & Mukherjee, 2021*; *Yadav & Vishwakarma, 2020*) pertains to the process of extraction and understanding of sentiments and viewpoints conveyed in text data.

Traditional methods of understanding sentiments often struggle to grasp the meanings found in reviews (*Bensoltane & Zaki, 2023*; *Yang, Gao & Fu, 2021*). This is especially true when it comes to aspect based sentiment analysis (ABSA) which focuses on examining emotions tied to aspects or features of products or services. The traditional methods used for ABSA often struggle to interpret the true meanings of words associated with those aspects. This process of knowing the true sense related to a word used in a text is commonly known as the word sense disambiguation (WSD). WSD can be explained by this example. For example, let us consider the word 'predictable'. If the word 'predictable' is being used in a review of smartphones while mentioning the affordability of smartphone functions, it conveys a positive sentiment. However, if the same word is used in the review of a movie while mentioning the predictable plot of that movie, it carries a negative connotation. Therefore, if a sentiment analysis system fails to differentiate between these nuances of the word 'predictable', it could lead to catastrophic results. Within the literature, there has always been a research gap in effectively incorporating WSD techniques into sentiment analysis models (*Missen, Boughanem & Cabanac, 2009*). This study addresses this challenge by integrating WSD with the ABSA framework. This provides an opportunity to surmount this obstacle and enable an understanding of feelings expressed towards aspects.

### Research gap

Many researchers have been proposing WSD techniques in general and some have also explored such techniques in the context of traditional sentiment analysis. However, the number of such attempts remains scarce when it comes to ABSA. This scarcity is particularly noteworthy given the heightened significance of ABSA, which can directly influence business revenues, especially if we are dealing with some product or service reviews. In classical sentiment analysis, the focus is on the overall sentiment expressed in a piece of opinion, while ABSA endeavors to dissect sentiments at a granular level, delving

into the sentiments expressed towards individual aspects of a product, service, or experience. Consequently, the accurate disambiguation of word senses becomes even more crucial in ABSA, as the misinterpretation of ambiguous terms can significantly impact the analysis results. Despite the acknowledged importance of word sense disambiguation in ABSA, only a select few researchers have ventured into this domain. We highlight two notable attempts from the literature that have proposed WSD approaches for ABSA. *Farooq et al. (2015)* proposed a WSD approach that involves building a lexicon and disambiguation of local context. The lexicon building involves five major steps, which also constitute extraction of product reviews from external data sources like eBay, Amazon, *etc*. The results demonstrated that the WSD approach proposed improved the ABSA results. However, the proposed approach is cumbersome and involves the usage of too many external sources hence making the real-time implementation of such an approach impossible. In *Grissette & Nfaoui (2022)*, authors propose the development of neural sense disambiguation (NSD) in two steps: (1) biomedical concept-based disambiguation and (2) universal word sense disambiguation. The major shortcomings of this work are that the work is solely limited to the biomedical domain and the proposed WSD approach is a semi-supervised approach. Both these attempts summarize the nature of the complexity involved in the WSD task, hence making researchers indulge in the proposal of complicated and supervised solutions. This highlights a notable research gap and highlights the untapped potential for advancements in ABSA methodologies. By incorporating robust word sense disambiguation techniques into ABSA frameworks, researchers can enhance the accuracy and depth of sentiment analysis in complex linguistic contexts, thereby facilitating more informed decision-making processes for businesses and stakeholders.

To overcome these limitations, this research article proposes a pioneering approach to WSD for ABSA by integrating the power of WordNet and the famous transformer architecture bidirectional encoder representations from transformers *i.e.*, BERT (*Vaswani et al., 2017*). BERT comes with some limitations and the core limitation of BERT is that while it excels at generating context-aware embeddings, it does not explicitly resolve polysemy. In polysemy (*Gries, 2015*), a word may have multiple senses depending on the context. This limitation is particularly problematic in sentiment analysis in general and ABSA in particular. For instance, BERT may capture the overall context of a review, but it does not disambiguate between different meanings of a word such as "sharp" in "sharp criticism" (negative) *vs.* "sharp knife" (positive). It may lead us to a situation where we have associated a wrong semantic orientation with a particular product or service feature or aspect. In this work, we particularly focus on this issue by integrating WSD with BERT. Unlike previous approaches that rely solely on the broad contextual understanding provided by transformer models, our method introduces an additional layer of precision by disambiguating the sense of words before leveraging BERT's contextual representations. This allows for more accurate sentiment classification at the aspect level, where ambiguity can drastically affect the outcomes. By transforming a simple review into a graph structure, the integration of contextual information and modelling intricate relationships between

different aspects is facilitated by graph convolutional networks (GCNs). This approach facilitates a more comprehensive and nuanced understanding of customer sentiments. In this work, we aim to demonstrate the effectiveness of combining BiLSTM and GCN in capturing inter-dependencies among various aspects of hotel reviews. The proposed architecture, which combines both BiLSTM and GCN, along with another strong combination of WSD and BERT, is set to improve the ABSA research. The implications of this research have the potential to significantly influence decision-making processes within the hospitality sector, ultimately leading to enhanced customer satisfaction and improved service delivery.

This article is organized as follows: "Contributions" lists the major contributions of this work. In "Related Work", we describe the related work of this article. "Model Explanation" describes the proposed neural network architecture in detail, and in "Experiments", we provide details of the experiments conducted.

## RELATED WORK

The use of deep learning models, in the context of graph-based structures has driven significant progress in ABSA. This section offers a summary of advancements in ABSA that highlight cutting-edge models and their exceptional performance across benchmark datasets (*Veyseh et al., 2020*; *Cai et al., 2020*; *Wang et al., 2020*; *Chen, Tian & Song, 2020*; *Ali Awan et al., 2021*). Subsequent sub-sections delve into the research articles categorized based on the methodologies employed and their relevance, to our subject matter.

### Word sense disambiguation approaches in ABSA

From the perspective of WSD only a small number of individuals have attempted to tackle the ABSA concern. We are discussing two major approaches proposed in this regard. *Farooq et al. (2015)* introduced a method for WSD that blends forming a lexicon with disambiguating context. The process involved five stages in constructing the vocabulary, including collecting reviews from platforms such as Amazon and eBay. Their findings showed that using this WSD technique enhanced results, in ABSA. However, the approach is deemed intricate for use and depends heavily on external references.

*Grissette & Nfaoui (2022)* proposed a neural sense disambiguation (NSD) framework, in a study. The framework was split into two components: disambiguation focused on concepts, and disambiguation centered around general word sense distinctions. Considered promising with its application of supervised methods, for WSD this approach is primarily tailored to the field of biomedicine. These studies highlight the challenges associated with WSD tasks and the tendency for researchers to lean toward solutions. Breakthrough possibilities, in the field of ABSA are highlighted by the research gap that has been uncovered.

### Graph-based approaches in ABSA

Recent research has put forward graph-based techniques to enhance the accuracy of ABSA. In one of the studies, by *Zhang, Zheng & Yang*'s *(2024)* work they explored the use of

dependency graphs in conjunction with GCNs. Their article introduces SD GCN as an approach, to ABSA that merges GCNs with dependency graphs to analyze aspects and sentiments more effectively by leveraging biaffine attention and GCNs to capture distant syntactic relationships comprehensively.

A new approach, for ABSA known as DSSK-GAN by *Liu et al. (2024)* was introduced in 2024 to capture aspect sentiment relationships through the integration of both dynamic external knowledge graphs. DSSK-GAN merges knowledge graph data with semantic representations to enhance sentiment analysis capabilities effectively and robustly based on results, from experiments conducted using ABSA datasets.

*Veyseh et al. (2020)* presented a learning model (*Kajla et al., 2020*, *2021*) that relies on personalized vectors and importance scores derived from dependency trees for performance citing. In the year (*Cai et al., 2020*) introduced the hierarchical graph convolutional network (Hier GCN) which outperformed models, on various standard datasets citing.

## Syntactic information and semantic correlations

In their work published by *Li et al. (2021)* unveiled a method called DualGCN that not only considers sentence structure but also looks at how different words are connected to address key issues in ABSA. *Liu et al. (2021)* utilized graph networks in GCN to recognize relationships between words, while *Awan et al. (2021)* on the other hand, used bidirectional attention to pinpoint specific elements within a given text.

## Knowledge integration for enhanced ABSA

*Zhao & Yu (2021)* introduced a technique that enhances ABSA using a combination of a sentiment knowledge graph (SKG) and a pre-trained BERT model, for capturing aspect sentiment interactions. Their model surpasses simpler ABSA methods. Another fascinating study by *Liang et al. (2022)*, introduces SenticGCN. A graph network tailored for aspect-based sentiment analysis. SenticGCNs improve dependency graphs by adding sentiment details related to aspects through the incorporation of knowledge, from SenticNet—an approach that differs from previous methods that primarily use dependency tree data exclusively for analysis purposes. Experimental results on datasets indicate that SenticGCNs surpass state-of-the-art techniques, in achieving highly detailed sentiment analysis.

## Attention mechanism, transformers and beyond for improved ABSA

*Wang et al. (2020)* introduced a model named RGAT that has shown performance compared to previous attention-based models on standard datasets. The findings have revealed the limitations of existing ABSA models while highlighting the potential for RGAT to significantly enhance ABSA. In 2020, *Chen, Tian & Song (2020)* presented directional graph convolutional networks (DG-CNN), an approach for ABSA that leverages the connections within a sentence's dependency structure.

ABSA has advanced over time with models shifting focus towards identifying aspects and adapting to various domains more effectively. Traditional methods like ATAE-LSTM and flan-t5-large-ABSA face challenges in extending their applicability beyond domains such as restaurant and hotel reviews due to their reliance on attention mechanisms. Although more sophisticated models like DeBERTa and GPT4-Turbo improve their understanding in these areas based on the context but there are still many constraints faced by both these models (*Mughal et al., 2024*).

UrduAspectNet (*Aziz et al., 2024*), utilize attention and dependency parsing to address language challenges while ASHGAT (*Ouyang et al., 2024*) employs hypergraph attention to enhance aspect relationship modeling instead. However, the computational requirements of both models pose implementation challenges. Based on research by *Aziz et al. (2024)*, *Ouyang et al. (2024)*, BERT-ASC (*Murtadha et al., 2024*) employs sentence construction for enhanced aspect recognition, but this approach depends on rephrasing strategies. Optimization of these models remains challenging; however, hybrid approaches such as TextGT (*Yin & Zhong, 2024*) that merge GNN and Transformer structures show potential in harmonizing semantic data. The importance of effective ABSA models of accommodating different languages and fields is underscored by these innovations.

## ABSA work focusing on hotel reviews

Many research studies have focused on analyzing hotel reviews to learn more about what customers prefer on sites like TripAdvisor (www.tripadvisor.com) and Booking (www.booking.com). *Ghosal & Jain (2023)*, for instance, developed a specialized sentiment analysis algorithm for the tourism sector to better understand customer feedback. Similarly, *Li et al. (2023)* examined how aspects such as service quality and taste influence their business strategies. In another similar work for hotel review analysis published in 2023, *Tayal, Yadav & Arora (2023)* introduced a multi-criteria decision making (MCDM) approach for ranking the products. It used ABSA approaches for extracting customer opinions and integrated them with customer preferences (*Tayal, Yadav & Arora, 2023*). *Alqaryouti et al. (2024)* explored the assessment of government applications by employing a mix of lexicon-driven and rule-based methods to grasp nuanced emotions in order to improve user satisfaction. In another work, *Mehra (2023)* looked at how emotions (especially surprise emotion) play a part in tourism and discovered that it has a big influence on how visitors intend to behave. *Vassilikopoulou, Kamenidou & Priporas (2024)* focused on enhancing customer service by examining Airbnb feedback to identify concerns. In a work presented in *Doan et al. (2024)*, a comprehensive dataset for evaluating sentiment in the hospitality sector has been introduced recently in 2024 and is known as HOSSemEval-EB23. It enables several sentiment analysis tasks. *Nawawi et al. (2024)* utilize zero-shot learning (ZSL) to assess tourism feedback and suggest improvements for enhancing visitor experiences by employing ABSA techniques. In a study published in 2024, *Iswari et al. (2024)* delve into enhancing the administration of tourism by elevating the precision of sentiment analysis through leveraging similarity when evaluating feedback from tourists.

## Diverse approaches and comparative insights

Many research articles have delved into the realm of ABSA over the years by focusing on different related challenges. To enhance performance across datasets, *Zhang & Qian (2020)* introduced a framework leveraging word cohesiveness and varied syntactical structures. These studies offer comparisons that shed light on both the capabilities and constraints of models.

Summarizing the Related Work section, it can be said that the use of graph-based models, attention mechanisms and knowledge integration have played a significant role in the advancements of ABSA. Analyzing the proposed models shows how the ABSA research field is evolving, with each new model having its unique strengths. These developments suggest possibilities for improving sentiment analysis applications, in industries and disciplines. The comparative analysis of the proposed models reveals the changing landscape of ABSA research, as each model possesses distinct strengths. These advancements demonstrate potential for enhanced sentiment analysis applications in a variety of fields (*Veyseh et al., 2020*; *Cai et al., 2020*; *Wang et al., 2020*; *Chen, Tian & Song, 2020*; *Li et al., 2021*; *Liu et al., 2021*; *Zhao & Yu, 2021*; *Zhang & Qian, 2020*; *Missen, Boughanem & Cabanac, 2010*). Nevertheless, there is still a significant amount of work to be done in the semantic aspects of sentiment analysis, and the work proposed in this article can be regarded as another step in this direction.

### Research questions

After going through the related work, several gaps and opportunities for further research have been identified that are listed below:

- How can the integration of WSD with BERT address the limitations of contextual ambiguity in aspect-based sentiment analysis, and to what extent does this integration improve sentiment classification performance at the aspect level?
- How to improve the existing annotation scheme of ABSA data collections?
- Will the idea of using BERT Embeddings in WSD help in identifying the real sense of the words in the input sentence?
- Could different similarity measures in WSD impact the overall outcomes of ABSA?

In the next sections, we will find answers to these research questions.

## Contributions

Below, we outline the key contributions of this work, which primarily address the research gaps identified in the existing literature.

- We propose a novel word sense disambiguation technique that improves the results of ABSA. The proposed WSD technique is computationally less complex and efficient.
- We propose a hybrid neural network architecture that combines the strengths of WSD, BiLSTM and GCN for ABSA.
- We demonstrate a multi-modal learning approach that incorporates both sequential and structural information for ABSA.

- We prepare a data collection named RABSA (refined aspect-based sentiment analysis) dataset. RABSA is a far more refined dataset as far as information labeled is concerned.
- We evaluate the proposed model on much RABSA dataset to demonstrate its effectiveness and compare its performance against existing state-of-the-art methods.
- We compare the performance of different similarity metrics (*i.e.*, cosine similarity, Manhattan distance, and Euclidean distance) on ABSA results.

## THE PROPOSED METHODOLOGY

### Model explanation

Let $S$ denote a given sentence extracted from a hotel review, and $A$ represent a specific aspect within the context of the review. In ABSA, an "aspect" refers to a specific component or feature of a product or service being discussed in a review. For example, in a hotel review, aspects might include "room cleanliness," "staff behavior," or "location." Accurately identifying sentiments toward each aspect allows for more granular sentiment analysis, enabling a deeper understanding of customer feedback.

The primary objective is to formulate a model that predicts the semantic orientation $SO$ of the aspect $A$ based on the information conveyed in the provided sentence $S$. Mathematically, this can be expressed as the conditional probability:

$$P(SO(A)|S) \tag{1}$$

This probability signifies the likelihood of the semantic orientation of aspect $A$ conditioned on the content of the given sentence $A$. The purpose is to build a predictive model that can predict the semantic orientation of a given aspect in a hotel review. In this section, we explain this proposed hybrid neural network architecture, which is shown in Fig. 1.

### Overview

Figure 1 depicts our innovative architecture for aspect-based sentiment analysis, characterized by the incorporation of sophisticated methodologies of BERT, GCN, BiLSTM, and a WSD module. The proposed approach utilizes BERT embeddings for the word sense disambiguation module, which dynamically identifies the most contextually pertinent WordNet examples for a specific word sense. This word sense disambiguation module is a significant contribution to this effort. Furthermore, we employ an attention layer subsequent to the BiLSTM to emphasize the importance of particular WordNet instances during processing. This attention mechanism allows the model to prioritize contextually significant instances, facilitating a more refined interpretation of sentiment related to a specific component. The entire architecture of the proposed model, featuring a significant contribution from the WSD module, seeks to enhance ABSA outcomes.

### Word sense disambiguation

WSD (*Bevilacqua et al., 2021*; *Vidhu Bhala & Abirami, 2014*) is essential for ABSA as it addresses the intrinsic ambiguity of polysemous words. The primary objective of WSD is

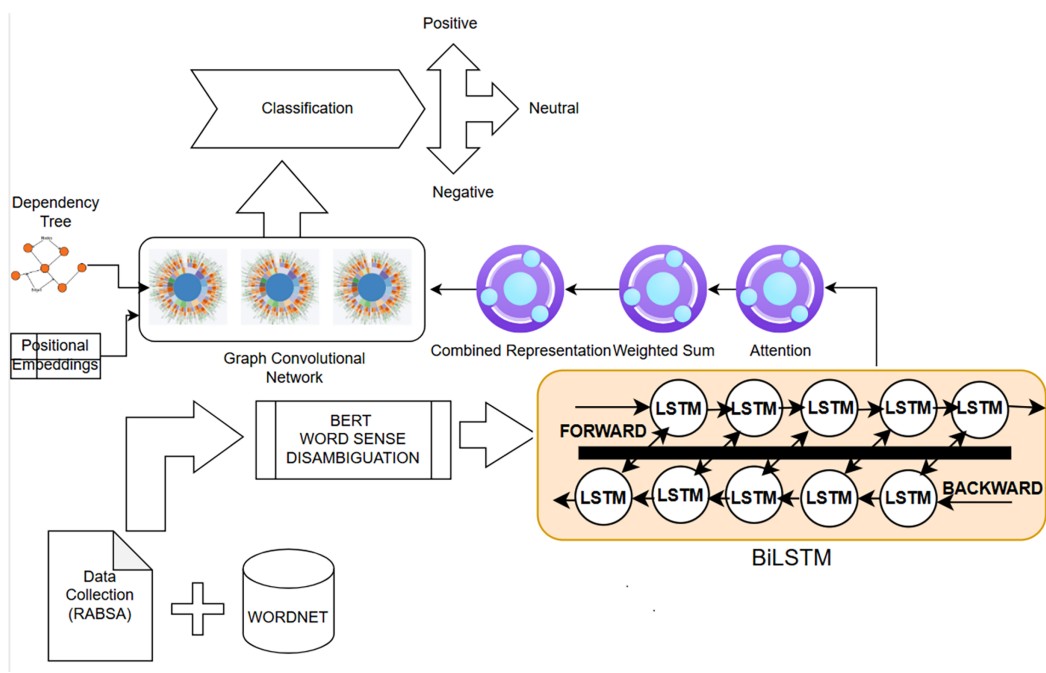

**Figure 1** The architectural diagram of the proposed system.

to ascertain the accurate meaning within a certain context. ABSA seeks to evaluate feelings toward particular characteristics or entities in text, and precise disambiguation of word senses improves the accuracy of sentiment predictions. In the absence of WSD, the likelihood of sentiment misinterpretation escalates, resulting in diminished accuracy in identifying elements and their corresponding attitudes. Incorporating WSD into ABSA models aids in addressing polysemy, diminishing ambiguity, and utilizing external knowledge resources, hence enhancing overall performance and facilitating nuanced sentiment analysis about various aspects of natural language text.

This study introduces an innovative method for word sense disambiguation utilizing BERT embeddings in conjunction with WordNet, a widely recognized lexical database for the English language. The primary concept entails producing BERT embeddings for a specific word in a phrase and juxtaposing them with embeddings from illustrative sentences obtained from WordNet (see Figs. 2 and 3).

### WordNet

WordNet (*Miller, 1995*) is a lexical database and semantic network that provides significant support for interpreting how words in the English language are related. It groups words into synsets (synonymous word sets) and provides extensive information for each entry. That will mean detailed statistics for all words: synonyms, part-of-speech information including definitions and hierarchical relationships *etc*. Figure 4 illustrates an example of the information provided for the word "dirty".

WordNet is a quintessential resource for different natural language processing applications, aiding in better comprehension and exploration of the English language

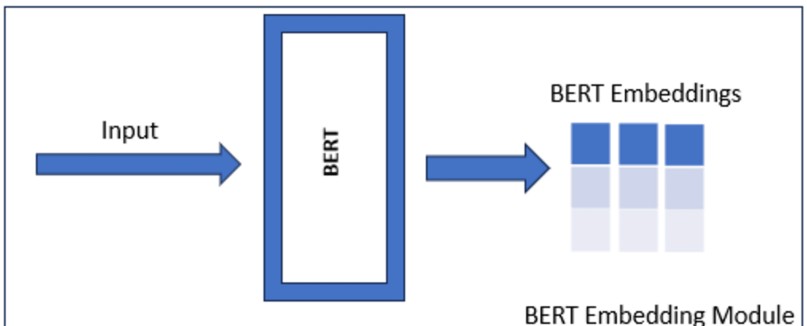

**Figure 2 BERT embeddings module.**

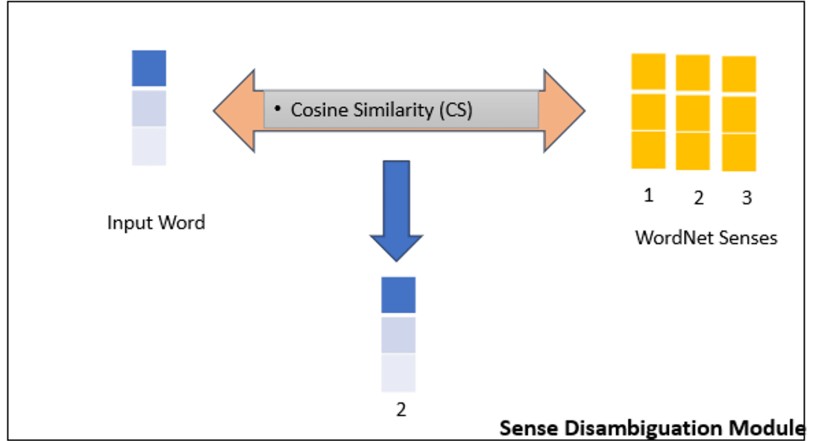

**Figure 3 Word sense disambiguation module explained.**

(*Zhang, Nath & Mazzaccara, 2023*; *Poomagal et al., 2023*). A brief introduction to the various aspects WordNet offers for each word is given below:

- Synonyms and synsets: WordNet groups words into synsets, which are sets of synonyms that represent a specific concept. For example, the synset for "dog" may include words like "pooch," "canine," and "hound".
- Part of speech: Each word entry in WordNet is labeled with its part of speech, such as noun, verb, adjective, or adverb. This information helps users understand the grammatical context in which a word is typically used.
- Definitions (Gloss): WordNet provides concise and informative glosses or definitions for each word. These glosses offer quick insights into the meaning of a word, aiding users in grasping its core concept.
- Hypernyms and hyponyms: WordNet establishes hierarchical relationships between words. A hypernym is a more general term, while a hyponym is a more specific term. For example, "animal" is a hypernym of "dog," and "dog" is a hyponym of "animal".

**Verb**

- S: (v) **dirty**, soil, begrime, grime, colly, bemire (make soiled, filthy, or dirty) *"don't soil your clothes when you play outside!"*

**Adjective**

- S: (adj) **dirty**, soiled, unclean (soiled or likely to soil with dirt or grime) *"dirty unswept sidewalks"; "a child in dirty overalls"; "dirty slums"; "piles of dirty dishes"; "put his dirty feet on the clean sheet"; "wore an unclean shirt"; "mining is a dirty job"; "Cinderella did the dirty work while her sisters preened themselves"*
- S: (adj) **dirty** ((of behavior or especially language) characterized by obscenity or indecency) *"dirty words"; "a dirty old man"; "dirty books and movies"; "boys telling dirty jokes"; "has a dirty mouth"*
- S: (adj) **dirty**, filthy, lousy (vile; despicable) *"a dirty (or lousy) trick"; "a filthy traitor"*
- S: (adj) **dirty**, contaminating (spreading pollution or contamination; especially radioactive contamination) *"the air near the foundry was always dirty"; "a dirty bomb releases enormous amounts of long-lived radioactive fallout"*
- S: (adj) **dirty**, pestiferous (contaminated with infecting organisms) *"dirty wounds"; "obliged to go into infected rooms"- Jane Austen*
- S: (adj) **dirty**, dingy, muddied, muddy ((of color) discolored by impurities; not bright and clear) *"dirty" is often used in combination; "a dirty (or dingy) white"; "the muddied grey of the sea"; "muddy colors"; "dirty-green walls"; "dirty-blonde hair"*
- S: (adj) **dirty**, foul, marked-up ((of a manuscript) defaced with changes) *"foul (or dirty) copy"*
- S: (adj) **dirty**, ill-gotten (obtained illegally or by improper means) *"dirty money"; "ill-gotten gains"*
- S: (adj) **dirty** (expressing or revealing hostility or dislike) *"dirty looks"*
- S: (adj) cheating, **dirty**, foul, unsporting, unsportsmanlike (violating accepted standards or rules) *"a dirty fighter"; "used foul means to gain power"; "a nasty unsporting serve"; "fined for unsportsmanlike behavior"*
- S: (adj) **dirty**, sordid, shoddy (unethical or dishonest) *"dirty police officers"; "a sordid political campaign"; "shoddy business practices"*
- S: (adj) **dirty** (unpleasantly stormy) *"there's dirty weather in the offing"*

**Figure 4 WordNet excerpt for word "dirty".**

- Meronyms and holonyms: WordNet indicates part-whole relationships. Meronyms are parts of a whole, and holonyms are the wholes. For instance, "finger" is a meronym of "hand," and "hand" is a holonym of "finger".
- Antonyms: WordNet includes antonyms for many words, highlighting opposites. For example, the antonym of "happy" might be "sad".
- Usage examples: WordNet often provides sample sentences or phrases to illustrate the usage of a word in context. This aids users in understanding how a word is typically employed.
- Derivationally related forms: WordNet includes information on words that are derived from the entry, such as different verb forms, adjectives, or adverbs.

*Word sense selection*

The proposed method (Fig. 2) for word sense disambiguation capitalizes on BERT's ability to capture contextual nuances, providing a robust means of discerning the sense in which the word is used in a specific context. The research explores the integration of external

knowledge from WordNet to enhance the model's understanding of diverse word meanings, leading to improved accuracy in word sense disambiguation. Experimental results demonstrate the effectiveness of the proposed approach in capturing nuanced word senses, making it a valuable contribution to the field of natural language processing. When comparing BERT embeddings of an input sentence and examples, we measure the similarity using the most effective method in literature, *i.e.*, cosine similarity. Cosine similarity measures the cosine of the angle between two vectors in an embedding space. This measure is particularly effective in determining how closely related two embeddings are, with values closer to 1 indicating higher similarity. By applying cosine similarity between a target word's contextual embedding and example sense embeddings from WordNet, the model selects the sense that best matches the intended meaning within the given context.

The cosine similarity is computed between the BERT embeddings of the input sentence and each example sentence. It ranges from −1 (completely dissimilar) to 1 (identical). At the end, we select the sense with the highest cosine similarity with the input sentence.

Let $S_1, S_2, \ldots, S_n$ be the WordNet senses of $W$. The selected sense $S_{\text{selected}}$ is determined by maximizing the cosine similarity:

$$S_{\text{selected}} = \arg\max_i(\text{Cosine Similarity}(E_W, E_{S_i})) \tag{2}$$

The whole process is also represented as an algorithm in Algorithm 1. We use a pre-trained uncased version of the BERT base model (768 dimensions per embedding) for generating BERT embeddings.

### Demonstration

Our model effectively handles ambiguous terms by leveraging WSD combined with context-aware embeddings. Let us consider an example for demonstration, consider the term "bank," which has multiple meanings depending on the context.

**Sentence 1:** "I deposited my paycheck at the bank." Predicted sense: bank (financial institution). Context embedding: The model identifies the context as related to finances and predicts the correct sense of the word "bank." Prediction: Positive sentiment towards banking services. **Sentence 2:** "We walked along the bank of the river." **Predicted Sense:** bank (riverbank). Context embedding: Here, the model recognizes the word "bank" refers to a natural landform adjacent to a river, disambiguating correctly based on the surrounding context. Prediction: Neutral sentiment regarding location. Performance insights: Traditional models (*e.g.*, VSM, random forest, and naive Bayes baselines) often rely solely on bag-of-words or surface-level features, leading to incorrect predictions for sentences with ambiguous terms. For instance, a baseline model might confuse "bank" in both sentences, as it would treat the word in isolation without understanding the surrounding context. In contrast, our model's use of BERT embeddings, enhanced with WSD, resolves such ambiguity, improving sentiment prediction accuracy.

| Algorithm 1 | Algorithm for the selection of the "Real Word Sense". |
|---|---|
| 1: | **Input:** |
| 2: | $W$: Input word |
| 3: | $E_W$: BERT embedding of the input word |
| 4: | $Senses_W$: List of WordNet senses for the word $W$, represented as $\{S_1, S_2, \ldots, S_n\}$ |
| 5: | $Embeddings_W$: List of BERT embeddings for the WordNet senses, represented as $\{E_{S_1}, E_{S_2}, \ldots, E_{S_n}\}$ |
| 6: | **Output:** |
| 7: | $SelectedSense$: The WordNet sense with the highest cosine similarity |
| 8: | **Algorithm:** |
| 9: | Initialize MaxSimilarity $= -\infty$ |
| 10: | **for** $i = 1$ to $n$ **do** |
| 11: | Compute $CosineSimilarity_i =$ Cosine Similarity$(E_W, Embeddings_W[i])$ |
| 12: | **if** $CosineSimilarity_i >$ MaxSimilarity **then** |
| 13: | Update MaxSimilarity $= CosineSimilarity_i$ |
| 14: | Update $SelectedSense = Senses_W[i]$ |
| 15: | **end if** |
| 16: | **end for** |
| 17: | **Output** $SelectedSense = 0$ |

## Methodological summary

### 1. Preprocessing and data preparation

The initial step involved tokenizing sentences into individual words and filtering for specific parts of speech, specifically verbs, adjectives, and adverbs, as these words often carry the core sentiment of the text. POS tagging is performed allowing us to retain only the words relevant for sentiment analysis.

### 2. Word sense disambiguation

Each word identified in the previous step was mapped to its possible senses using WordNet. For each word, we retrieved all possible senses and their glosses. To determine the appropriate sense in context, cosine similarity was computed between the BERT embedding of the word in context and the embeddings of each sense definition in WordNet. The sense with the highest similarity score above a set threshold is selected as the disambiguated sense of the word. This similarity threshold is optimized based on validation data, allowing a balance between accuracy and computational efficiency.

### 3. BERT embeddings integration

We use a pre-trained BERT model (BERT-base from the Hugging Face Transformers library) to generate contextual embeddings for each word within a sentence. Each word's embedding represented its meaning in the sentence context, capturing both syntactic and

semantic nuances. For each word's potential sense from WordNet, a similarity comparison is conducted with its contextual BERT embedding, enabling us to select the sense most aligned with the word's contextual meaning.

4. **Aspect-based sentiment analysis**

After obtaining the sense-disambiguated representations of words, we focus on identifying aspects within the sentences using dependency parsing. Each aspect is represented by its BERT-enhanced sense embedding, combined with the surrounding context through an attention mechanism. This attention-enhanced representation is then fed into a GCN for sentiment classification. The GCN incorporated both the syntactic dependency structure and the contextual information, capturing both the relationships among aspects and their associated sentiments. This is followed by a sentiment classification layer, categorizing each aspect as positive, negative, or neutral.

5. **Integration challenges and parameter tuning**

Parameter tuning is conducted to optimize the similarity threshold for sense selection and the hyperparameters of the sentiment classification model. Hyperparameters, such as embedding dimensions, learning rate, and regularization terms, were fine-tuned through grid search on a validation dataset. Notably, challenges are encountered in cases where multiple senses had similar similarity scores, and further refinement was necessary for optimal disambiguation.

## Experiments

In this section, we explain how we collect our data collection, its annotation details, and its structural details, and compare it with existing annotation schemes. We also reveal experimental details, including results and discussions.

## Data collection

### Data acquisition

We manually collected reviews (positive, negative, and neutral) from two popular online hotel booking sites, *i.e.*, www.booking.com and www.tripadvisor.com. In the start, we wrote a Python script for scraping the reviews from these websites. However, it was observed that lots of important data were being lost during the process, and also, some useless data was being obtained in the results; hence, this idea was dropped, and we consulted the manual collection of data. A total of 1,500 reviews from each category are collected (making 4,500 total reviews), which can further be divided into sentences and are ready for annotation.

During this process, some preprocessing steps are performed to normalize the reviews. This normalization includes removing any irrelevant content, such as URLs or promotional texts, and regulating punctuation and capitalization to maintain consistency across the dataset. This structured approach to data collection and preprocessing ensures that the dataset is consistent throughout and is ready for use for experimentation purposes.

### Data annotation

Annotation of all sentences is done by two experts. Both experts can understand, read, and write the English language very well. Both are students of MPhil English and are also aware of netiquettes. Both experts are provided annotation guidelines for annotating the data collection. Kappa measure score of annotation agreement turns out to be 0.81 which is deemed excellent in information retrieval tasks.

### Structure of data collection

As mentioned above, we prepared our own data collection (known as "RABSA" data collection referring to "refined aspect-based sentiment analysis" data collection) using reviews from two online hotel booking sites, *i.e.*, www.booking.com and www.tripadvisor.com. In Fig. 5, an excerpt from the labeled data collection RABSA is provided. The data is arranged in XML format and represents a set of sentences, each encapsulating a sentiment analysis annotation (*Saad Missen et al., 2018*). Specifically, it includes information about the sentiment polarity of individual sentences along with aspects within each sentence that contribute to the expressed sentiment.

Each <sentence> element is identified by a unique identifier (sid) and is associated with a review identifier (rid) which is the identity of the review from which the sentence has been extracted. The sentiment polarity is indicated by the "polarity" attribute within the <sentence> element, and it can be either "positive," "negative," or another relevant sentiment label. The main content of each sentence is enclosed within the <text> element, providing the actual textual content of the sentence. Aspects contributing to the sentiment are specified within the <aspects> section of each sentence. Each <aspect> element includes details such as the term indicating the aspect (term), its position in the sentence (position), a related term if applicable (RelTerm), the category of the aspect (category), and the polarity of the aspect (polarity). In the given example, the focus is on aspects like "staff," "rooms," and "cleanliness," and all of them are associated with a positive sentiment. This XML format is designed to capture sentiment information at both the sentence and aspect levels, facilitating detailed sentiment analysis of the provided textual data.

The data in Fig. 6 represents the data annotated according to the template of the SemEval evaluation campaign. Comparing RABSA with the labeled data collection of Fig. 6, it can be observed that RABSA is far more refined as far as information labeled is concerned. RABSA not only labels the polarity of the aspects but also the polarity of the sentence, which can eventually lead to determining the polarity of the overall review. In addition, RABSA is simpler in its annotation. RABSA abolishes the <aspectCategories> element from its annotation and makes it part of the <aspect> element as an attribute. RABSA also simplifies the positioning of aspect terms by replacing character positions with word positions, which makes the process of labeling and computation much easier.

A figurative comparison of both data collections, *i.e.*, RABSA and SemEval, is also given in Table 1 where a category-wise comparison for all three categories is provided.

```
<sentences>

<sentence sid="AS813" rid="r01" polarity="positive">

<text>Staff, rooms and cleanliness were all good</text>

<aspects>

<aspect term="staff" position="1" RelTerm="good" category="service" polarity="positive" />

<aspect term="rooms" position="2" Relterm="good" category="room" polarity="positive" />

<aspect term="cleanliness" position="4 " RelTerm="good "category="service" polarity="positive" />

</aspects>

</sentence>
</sentences>
```

**Figure 5** **Excerpt from the labeled data collection.**

### Ethical considerations

The review data used in this study was sourced from publicly available platforms, including Booking (www.booking.com) and TripAdvisor (www.tripadvisor.com), and has been collected in strict accordance with their terms of service. To ensure data privacy, all personal identifying information (PII) was anonymized, and no sensitive user data was included. Ethical guidelines were followed throughout the data collection process, ensuring compliance with data privacy standards.

## Hyperparameter settings

In the proposed architecture for aspect-based sentiment analysis, careful selection of hyperparameters is crucial for achieving optimal model performance. The principal hyperparameters are the learning rate, which dictates the step size in optimization, and the batch size, which affects the number of input samples processed per iteration. The quantity of graph convolutional layers and the hidden dimensions inside those layers in the GCN are essential in capturing complex interactions among elements. The dropout rate, utilized to alleviate overfitting, necessitates a precise adjustment to achieve an optimal equilibrium between model generalization and complexity. The BiLSTM layer is influenced by hyperparameter configurations, including the number of units and the dropout rate, which affect its capacity to capture sequential dependencies. Finally, the sentiment classification layer requires meticulous configuration, detailing the number of output units aligned with the sentiment classes and the activation function.

We commence the learning process for the proposed aspect-based sentiment analysis architecture by setting the starting learning rate to 0.001 when establishing the hyperparameters. This selection permits a moderate step size in optimization, with additional modifications possible depending on the convergence rate and stability seen during training. The batch size, denoting the quantity of input samples processed every iteration, is established within an initial range of 16 to 64. Increased batch sizes enhance computing efficiency; nonetheless, it is crucial to observe GPU memory limitations. The GCN is equipped with 1 to 2 layers, and the hidden dimensions within these layers range

```
<sentences>
 <sentence id="813">
  <text>All the appetizers and salads were fabulous, the steak was mouth watering
and the pasta was delicious!!!</text>
<aspectTerms>
 <aspectTerm term="appetizers" polarity="positive" from="8" to="18"/>
 <aspectTerm term="salads" polarity="positive" from="23" to="29"/>
 <aspectTerm term="steak" polarity="positive" from="49" to="54"/>
 <aspectTerm term="pasta" polarity="positive" from="82" to="87"/>
</aspectTerms>
<aspectCategories>
 <aspectCategory category="food" polarity="positive"/>
</aspectCategories>
 </sentence>
<sentences>
```

**Figure 6 Data as formatted in SemEval data collections.**

**Table 1 Figurative comparison of data collections _i.e._, RABSA _vs._ SemEval.**

| Sr. No | Data collection | Positive Training data | Test data | Negative Training data | Test data | Neutral Training data | Test data |
|---|---|---|---|---|---|---|---|
| 1 | SemEval 2014 (_Liang et al., 2022_) | 2,164 | 728 | 807 | 196 | 637 | 196 |
| 2 | RABSA | 3,000 | 1,000 | 3,000 | 1,000 | 1,000 | 500 |

from 64 to 256. These values are selected to capture intricate relationships among aspects of the graph structure. A dropout rate of 0.5 is initially applied in both the GCN and BiLSTM layers to mitigate overfitting, with the flexibility to adjust based on observed model generalization. The BiLSTM layer comprises 64 to 256 units to capture sequential patterns effectively. For the sentiment classification layer, the number of output units matches the sentiment classes (_e.g._, three for positive, negative, neutral), and the activation function is set to softmax, suitable for multi-class classification tasks. These initial hyperparameter settings serve as a baseline, and fine-tuning is conducted through systematic experimentation and performance evaluation.

For the ablation study, we re-ran the model by removing the WSD, GCN, and BiLSTM components individually while keeping the hyperparameters constant to examine each component's contribution to performance.

## Baselines

We employ the following baselines for performance comparison of the proposed system.

### Traditional machine learning algorithms

We experiment with traditional machine learning models such as support vector machines (SVM), random forests, and naive Bayes using our data collection by extracting the features.

### Simple neural networks

We compare the proposed architecture against simpler neural network architectures, such as a basic feedforward neural network or a model with LSTM (*Tang et al., 2015*) or with a single BiLSTM layer.

### Existing best results on ABSA

While we use a more enriched and different data collection for our work, we compare the performance of the proposed architecture with the existing best results (to the best of our knowledge) on SemEval14 restaurant data collection. (please refer to Tables 2–4).

- SenticGCN (*Liang et al., 2022*). This is a unique GCN contribution for ABSA which uses SenticNet to improve the effectiveness of knowledge graphs. By integrating contextual affective information with aspect-specific dependencies, SenticGCN outperformed previous results, demonstrating its effectiveness in fine-grained sentiment analysis tasks.
- SD-GCN (*Zhang, Zheng & Yang, 2024*). This is another interesting contribution for ABSA, which is based on a graph convolutional network approach. This work targets the limitations of attention-based and graph neural network methods by effectively capturing long-distance syntactic dependencies.
- MTABSA (*Zhao et al., 2023*). This work targets two sub-tasks related to ABSA *i.e.*, aspect term extraction (ATE) and aspect polarity classification (APC) by using a multitask learning model. This work has also used an attention mechanism for this purpose.
- T-GCN-BERT (*Tian, Chen & Song, 2021*). This contribution basically proposed another GCN-based work, which merges dependencies and types to build a graph, applies attention to weigh the edges and integrates layers to incorporate contextual data.
- R-GAT-BERT (*Wang et al., 2020*). They used a GAT that includes relational knowledge to deal with new dependency trees by rearranging and cutting down old ones.
- DualGCN (*Li et al., 2021*). DualGCN is an interesting work that proposed a dual graph convolutional network, *i.e.*, DualGCN. The major idea behind DualGCN is to combine syntactical structure and semantic correlations together.
- XLNET.XLNET is a major contribution by Google and Carnegie Mellon University researchers in 2019 (*Yang, 2019*). *Bashiri & Naderi (2024)* have summarized sentiment results for several transformers on several data collections. Among these, SemEval results relate to ABSA. Hence, the comparison is reported on the best results achieved on SemEval datasets.
- T5.T5 has been proposed by Google researchers in year 2019 (*Raffel et al., 2020*). *Bashiri & Naderi (2024)* have summarized sentiment results for several transformers on several data collections. Among these, SemEval results relate to ABSA. Hence, the comparison is reported on the best results achieved on SemEval datasets.

**Table 2 Performance comparison of the proposed approach with baselines (statistical significance denoted by $*p < 0.05$, $**p < 0.01$).**

| Model | Positive (%) | | Negative (%) | | Neutral (%) | |
|---|---|---|---|---|---|---|
| | Acc. | F1 | Acc. | F1 | Acc. | F1 |
| VSM | 81.00 | 78.03 | 80.13 | 78.43 | 76.39 | 69.89 |
| Random forest | 79.01 | 68.13 | 77.83 | 67.47 | 78.39 | 68.70 |
| Naive bayes | 79.01 | 74.38 | 74.83 | 68.43 | 73.19 | 68.10 |
| BiLSTM | 80.42 | 76.03 | 81.12 | 78.05 | 77.52 | 69.08 |
| GCN | 78.02 | 70.15 | 76.03 | 68.45 | 71.88 | 65.78 |
| SenticGCN (*Liang et al., 2022*) | 86.92 | 81.03 | 82.12 | 79.05 | 85.32 | 71.28 |
| Proposed | 90.52** | 85.03** | 89.12** | 86.05** | 84.32* | 71.06* |

# RESULTS AND DISCUSSIONS

## Evaluation metrics

The evaluation metrics selected for this study, *i.e.*, accuracy and F1-score, are commonly used in sentiment analysis to assess both the overall performance of classification models and their ability to correctly identify sentiment at a granular level. F1-score, in particular, is critical for imbalanced datasets, as it balances precision and recall. These metrics provide a comprehensive understanding of the model's performance in correctly classifying aspect-based sentiments.

## Results

Table 2 presents the performance metrics of various machine learning models, including VSM, random forest, naive Bayes, BiLSTM, GCN, a previous best model (as cited), and a newly proposed model. The metrics are segregated into positive, negative, and neutral categories with the corresponding accuracy (Acc.) and F1 scores in percentage. The proposed model outperforms all others with an accuracy of 90.52%, 89.12%, and 84.32% and F1 scores of 85.03%, 86.05%, and 71.06% in positive, negative, and neutral categories, respectively.

The rows in Table 2 represent the different models used for sentiment analysis. The first row represents the VSM, followed by random forest, naive Bayes, BiLSTM, GCN, several previous best models, and our proposed model. The table provides the accuracy and F1 scores of each model in the three sentiment categories. The proposed model outperforms all other models in all three categories. Table 3 shows the comparison of the best results of the proposed approach with the existing results reported. A separate table was needed for this comparison because the existing works listed in this table did not mention the category-wise results; hence making the category-wise results comparison infeasible.

## Comparison of BERT embedding similarity measures

This section introduces a comparative analysis of different similarity measures applied to BERT embeddings. The goal is to investigate and understand how various distance metrics

**Table 3 Comparison of the proposed approach with best overall reported results.**

| Model | Best reported | |
|---|---|---|
| | **Acc** | **F1** |
| LSTM (*Tang et al., 2015*) | 78.13 | 67.47 |
| T-GCN-BERT (*Tian, Chen & Song, 2021*) | 86.16 | 79.95 |
| R-GAT-BERT (*Wang et al., 2020*) | 86.60 | 81.35 |
| SD-GCN (*Zhang, Zheng & Yang, 2024*) | 88.14 | 81.42 |
| DualGCN (*Li et al., 2021*) | 84.27 | 78.08 |
| MTABSA (*Nazir et al., 2020*) | 86.88 | 81.16 |
| XLNET (*Yang, 2019*) | 85.95 | – |
| T5 (*Raffel et al., 2020*) | 87.47 | – |
| Proposed | 90.52 | 86.05 |

**Table 4 Impact of different similarity measures on overall results.**

| Similarity metrics | Positive (%) | | Negative (%) | | Neutral (%) | |
|---|---|---|---|---|---|---|
| | **Acc.** | **F1** | **Acc.** | **F1** | **Acc.** | **F1** |
| Cosine similarity | 90.52 | 85.03 | 89.12 | 86.05 | 84.32 | 71.06 |
| Euclidean distance | 88.10 | 84.01 | 86.98 | 85.00 | 81.12 | 69.77 |
| Manhattan distance | 88.85 | 85.23 | 86.02 | 86.11 | 81.34 | 69.76 |

affect the assessment of similarity between BERT embedding vectors. We specifically focus on the widely-used cosine similarity, Euclidean distance, and Manhattan distance.

### Cosine similarity

Cosine similarity is used to find the similarity between two vectors. The idea is to measure the cosine of the angle between any two vectors **A** and **B** if you want to find similarity between A and B. The formula of finding this cosine similarity ($\cos(\theta)$) is given below:

$$\text{cosine\_similarity}(\mathbf{A}, \mathbf{B}) = \frac{\mathbf{A} \cdot \mathbf{B}}{||\mathbf{A}|| \cdot ||\mathbf{B}||} \tag{3}$$

where **A.B** is the dot product between vectors **A** and **B** and $||\mathbf{A}||.||\mathbf{B}||$ represent the magnitude of both vectors.

### Euclidean distance

Euclidean distance between two points is actually the measure of the distance between them. The formula for finding the Euclidean distance between vectors **A** and **B** is given below:

$$\text{euclidean\_distance}(\mathbf{A}, \mathbf{B}) = \sqrt{\sum_{i=1}^{n} (\mathbf{A}_i - \mathbf{B}_i)^2} \tag{4}$$

### Manhattan distance

Manhattan distance is a metric used to find the sum of the absolute differences between the coordinates. The formula for finding the manhattan distance between vectors **A** and **B** is given below:

$$\text{manhattan\_distance}(\mathbf{A}, \mathbf{B}) = \sum_{i=1}^{n} |\mathbf{A}_i - \mathbf{B}_i| \tag{5}$$

The analysis of BERT embedding similarity measures comparison (as shown in Fig. 7) reveals notable performance distinctions. Cosine similarity consistently outperforms Euclidean and Manhattan distances across positive, negative, and neutral categories, achieving the highest accuracy and F1 scores. In particular, cosine similarity exhibits a significant advantage in the positive category with accuracy and F1 scores of 90.52% and 85.03%, respectively. Euclidean and Manhattan distances, while showing respectable performance, consistently fall short of the performance achieved by cosine similarity. The consistency in performance trends across categories suggests systematic impacts of these metrics.

### Receiver operating characteristic curve analysis

A receiver operating characteristic (ROC) curve is a popular way to visualize the performance of binary classifiers. Here we examine the ROC curves for three different classes: positive, negative and neutral. The curves in Fig. 8 represent the trade-off between true positive rate (TPR) and false positive rate (FPR), across different classification thresholds.

- Positive *vs*. rest (orange curve)—The orange curve corresponds to the positive class. It demonstrates exceptional predictive accuracy with an area under the curve (AUC) of 0.87. A high TPR (sensitivity) is achieved while maintaining a low FPR (specificity). This suggests that our model effectively identifies positive instances while minimizing false positives.
- Negative *vs*. rest (green curve)—The green curve represents the negative class. Its AUC is 0.84, indicating good performance. Although slightly less accurate than the Positive class, the negative curve still achieves a favourable balance between TPR and FPR.
- Neutral *vs*. rest (blue curve)—The blue curve corresponds to the neutral class. Its AUC is 0.77. While the neutral class exhibits reasonable discrimination, it lags behind the positive and negative classes in terms of accuracy.

It is very obvious from the ROC plot that the proposed model performs very well across all classes.

### WSD-BERT impact analysis

The WSD-BERT model exhibits substantial performance improvements in contexts characterized by word ambiguity. For instance, terms such as 'bank,' which may denote a

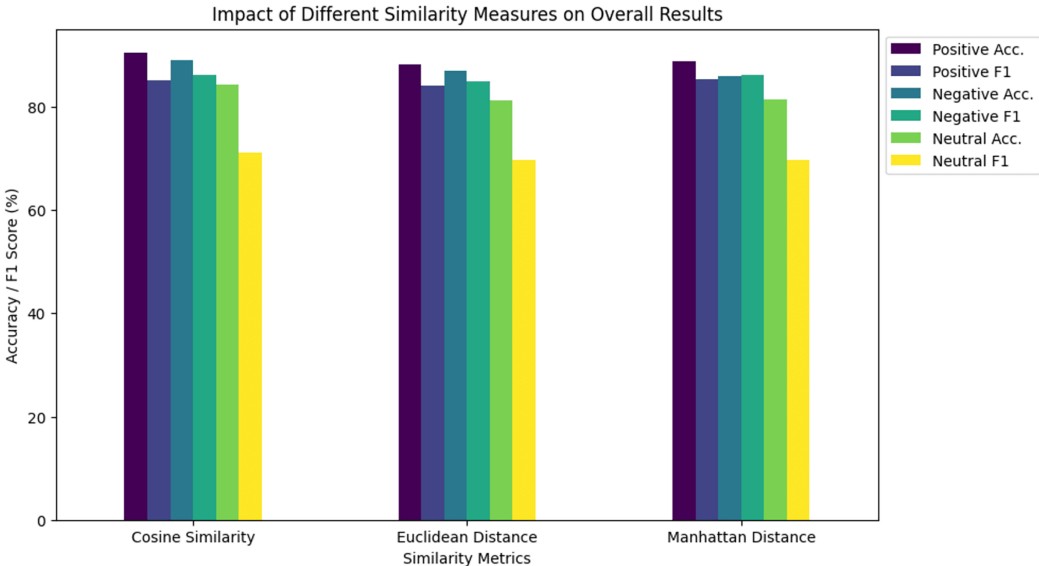

**Figure 7 Comparison of different similarity measures on overall results.**

financial organization or a riverbank, or 'sharp,' which can signify both favourable and unfavourable qualities, are effectively disambiguated by the suggested module, resulting in enhanced sentiment analysis accuracy. By clarifying these uncertainties, the model guarantees accurate identification of the sentiment linked to the aspect, which is particularly crucial for evaluations using complex or nuanced wording. Nonetheless, WSD does not consistently yield beneficial outcomes. In instances where the language is largely unambiguous, particularly when a term possesses a singular pertinent meaning inside the context, BERT's contextual embeddings are frequently adequate for accurate sentiment classification. In certain cases, implementing WSD may bring superfluous complexity, occasionally resulting in diminished performance. If a review phrase is clear and devoid of polysemous terms, the additional disambiguation may lead to overfitting or misclassification.

To address these constraints, future research could investigate dynamic methodologies that selectively implement WSD based on the degree of ambiguity present in the sentence. Moreover, including domain-specific knowledge in the WSD process or augmenting the granularity of lexical resources such as WordNet could further optimize performance in specialized domains.

In conclusion, our method of integrating WSD with BERT enhances the accuracy of ABSA and indicates a promising direction for sentiment analysis. The integration of WSD with BERT extends beyond ABSA, presenting opportunities for the advancement of state-of-the-art models, such as BERT, through the potential utilization of knowledge bases like WordNet. Future research may build upon this methodology, investigating how WSD can enhance model efficacy in additional domains requiring accurate semantic interpretation, thus providing a basis for wider applications in AI.

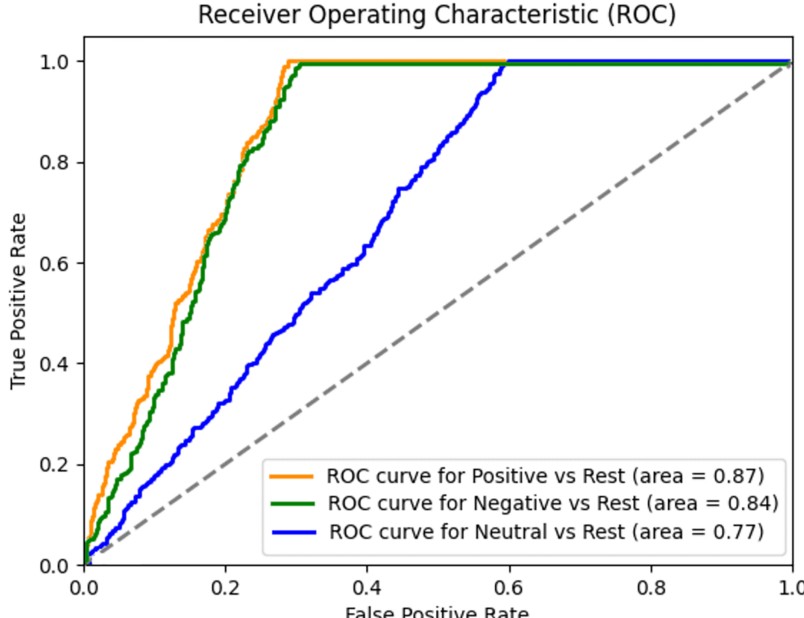

**Figure 8** ROC Curve for positive, negative and neutral classes-done on one-*vs*-rest method.

## Ablation study

In order to assess the impact of the proposed WSD and other modules on the overall performance of our sentiment analysis architecture, we conduct an ablation study. For checking the performance of a particular module, that module is extracted from the full model and its comparison is done with full proposed model. The results, as shown in Table 5, prove the specific contribution of the WSD module and other modules as compared to the overall effectiveness of the proposed model and provide insights into the necessity of incorporating WSD module in the context of aspect-based sentiment analysis for hotel reviews. It is to be noted that same dataset and same performance metrics are used for this study. All experiments are conducted using same the hyperparameter settings that can be seen in Table 6.

## LIMITATIONS AND POTENTIAL BIASES

- While the dataset was curated to represent diverse sentiments, there are potential domain-specific biases, as the reviews are sourced exclusively from the hospitality sector (hotels and travel platforms). This focus may limit the model's generalization to other domains. Future work could address this by including reviews from various domains to assess the broader applicability of the proposed approach.

- The impact of WSD-BERT is limited to data collections where there is ambiguity involved. While this cannot be considered a limitation because the purpose of this work is to target word sense ambiguity, it still limits the scope of the proposed approach. However, in the future, we aim to extend this work by integrating it with Schema.org.

**Table 5 Ablation study results.**

| Model | Positive (%) | | Negative (%) | | Neutral (%) | |
|---|---|---|---|---|---|---|
| | Acc. | F1 | Acc. | F1 | Acc. | F1 |
| Without WSD | 86.12 | 78.29 | 81.92 | 75.55 | 78.08 | 68.09 |
| Without attention | 85.52 | 79.83 | 80.52 | 74.57 | 78.69 | 67.77 |
| Without GCN | 87.12 | 82.88 | 86.79 | 78.08 | 82.14 | 76.17 |
| Without BiLSTM | 84.66 | 80.58 | 81.05 | 77.25 | 78.73 | 70.68 |
| Proposed | 90.52 | 85.03 | 89.12 | 86.05 | 84.32 | 71.06 |

**Table 6 Hyperparameter settings for aspect-based sentiment analysis.**

| Hyperparameter | Settings |
|---|---|
| Learning rate | 0.001 |
| Batch size | 16–64 |
| Graph convolutional layers | 1–2 |
| Graph convolutional layer hidden dimensions | 64–256 |
| Dropout rate (GCN, BiLSTM) | 0.5 |
| BiLSTM layer units | 64–256 |
| Sentiment classification output units | 3 |
| Sentiment classification activation function | Softmax |

# CONCLUSIONS

In this work, we made an effort to bridge one of the critical gaps in sentiment analysis by addressing the limitation of conventional ABSA methodologies in handling word sense ambiguities. This research introduced a novel approach to WSD by leveraging WordNet and the BERT transformer architecture. The addition of GCNs further enhanced the model's ability to understand intricate relationships among different aspects within hotel reviews, enabling structured representation through a graph-based framework. Our findings affirm that the proposed architecture—combining BiLSTM, GCN, and WSD— significantly improves sentiment analysis accuracy, directly addressing our research question by demonstrating how WSD integration helps resolve ambiguities that typically hinder ABSA performance. The ablation study further demonstrates the contribution of each component, particularly highlighting the WSD module's role in improving aspect-specific sentiment analysis. While the proposed approach can be very effective for the tourism and hospitality industry where the importance of a single word cannot be ignored, nothing stops its implementation in other domains as well. While promising, there remains scope for further enhancement. Future work could explore structured approaches, such as Schema.org, to enrich aspect-based sentiment analysis, providing even greater semantic depth.

### Funding

The authors received funding from the Deanship of Graduate Studies and Scientific Research at Najran University under the Growth Funding Program grant code (NU/GP/SERC/13/84). The funders had no role in study design, data collection and analysis, decision to publish, or preparation of the manuscript.

### Grant Disclosures

The following grant information was disclosed by the authors:
Deanship of Graduate Studies and Scientific Research at Najran University under the Growth Funding Program: NU/GP/SERC/13/84.

### Competing Interests

The authors declare that they have no competing interests.

### Author Contributions

- Asma Nadeem conceived and designed the experiments, analyzed the data, authored or reviewed drafts of the article, and approved the final draft.
- Malik Muhammad Saad Missen conceived and designed the experiments, analyzed the data, authored or reviewed drafts of the article, and approved the final draft.
- Mana Saleh Al Reshan conceived and designed the experiments, analyzed the data, authored or reviewed drafts of the article, and approved the final draft.
- Muhammad Ali Memon performed the experiments, prepared figures and/or tables, authored or reviewed drafts of the article, and approved the final draft.
- Yousef Asiri conceived and designed the experiments, analyzed the data, authored or reviewed drafts of the article, and approved the final draft.
- Muhammad Ali Nizamani performed the experiments, prepared figures and/or tables, authored or reviewed drafts of the article, and approved the final draft.
- Mohammad Alsulami performed the experiments, authored or reviewed drafts of the article, and approved the final draft.
- Asadullah Shaikh conceived and designed the experiments, analyzed the data, authored or reviewed drafts of the article, and approved the final draft.

### Data Availability

The data is available at GitHub and Zenodo:

- https://github.com/saadmissen/RABSA-Data-Collection.
- Malik Muhammad Saad Missen, M. (2024). RABSA: Aspect-Based Sentiment Analysis Data Collection for Hotel Reviews (Version 1). Zenodo. https://doi.org/10.5281/zenodo.11024326.

The code is available at GitHub and Zenodo: https://github.com/saadmissen/RABSA.

- Missen, M. M. S. (2024). RABSA-Data-Collection (II) [Data set]. Zenodo. https://doi.org/10.5281/zenodo.14302069.

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
