# Peer review of "Resolving ambiguity in natural language for enhancement of aspect-based sentiment analysis of hotel reviews"

_PeerJ Computer Science, doi:10.7717/peerj-cs.2635_

## Round 0.1 · original submission · Major Revisions

Dear authors,

Thank you for submitting your article. Feedback from the reviewers is now available. It is not recommended that your article be published in its current format. However, we strongly recommend that you address the issues raised by the reviewers, especially those related to readability, experimental design and validity, and resubmit your paper after making the necessary changes. Before submitting the paper, following should also be addressed:

1. Equations should be used with equation number. Explanation of the equations should be checked. All variables should be written in italics. Definitions and boundaries of all variables should be provided. Necessary references should also be given.
2. Many of the equations are part of the related sentences. Attention is needed for correct sentence formation.
3. All of the values for the parameters of all algorithms selected for comparison should be given.
4. Pros and cons of the methods should be clarified. What are the limitation(s) methodology(ies) adopted in this work? Please indicate practical advantages, and discuss research limitations.
5. Pay special attention to the usage of abbreviations. Spell out the full term at its first mention, indicate its abbreviation in parenthesis and use the abbreviation from then on.

Best wishes,

Reviewer 1 ·

Basic reporting

This article presents a new architecture for aspect-based sentiment analysis (ABSA) that stands out from others due to its innovative integration of cutting-edge techniques. The model utilizes BERT embeddings to capture precise aspect-specific meanings and introduces a novel word disambiguation module. This module dynamically selects the most relevant WordNet examples based on context, providing a deeper understanding of the intended sense and enhancing the model's semantic capabilities. Furthermore, the authors incorporate an attention layer after the BiLSTM to emphasize the importance of specific WordNet examples during processing. This attention mechanism allows the model to focus on contextually significant instances, enabling a more refined interpretation of sentiment associated with a particular aspect. The combination of word disambiguation and attention mechanisms sets our proposed architecture apart, offering an advanced framework for ABSA that is highly sensitive to contextual subtleties.

Experimental design

The authors have introduced a novel architecture that integrates multiple techniques to enhance the performance of Aspect-Based Sentiment Analysis (ABSA). The experimental design is sound, and the results indicate that the proposed method outperforms the baseline models. The research is a good fit for the aims and scope of this journal. The author has clearly defined the research questions, and the experiment was properly designed and conducted to address these questions.

Validity of the findings

The authors provide a detailed account of their experiments, ensuring the validity of their findings. Furthermore, they conduct an ablation study to assess the effectiveness of the integrated modules. A potential limitation of this study is that its originality may be compromised by the fact that each of the adopted models has been previously examined in the existing literature. Consequently, the primary value of this research lies in the novel integration of these models, rather than the introduction of new theoretical frameworks.

Additional comments

To improve the manuscript, I offer the following suggestions: Firstly, in the data collection section, it is essential to clarify how the authors collected and archived the 1500 samples from the two websites. Specifically, what percentage or number of comments were selected from each website, and how was the category determined? It appears that data annotation occurred after data collection; therefore, the authors should provide a clear explanation of their data collection process. Additionally, they should justify their selection of these two websites as data sources. I reviewed the source data website but cannot find out these information.

Secondly, the paper's presentation and organization could be enhanced. While I appreciate the detailed reporting in each section, some formulas could be condensed or relegated to an appendix. For instance, the cosine similarity formula is presented twice, which is unnecessary. In the baseline models section, it would be beneficial to provide a rationale for selecting these particular models for comparison. Furthermore, additional details about Table 3 are required, as it does not facilitate category-based comparisons; the authors should explain their criteria for selecting baseline models and why category-based comparison was not feasible or if they simply reported the best-reported results.

Lastly, since the proposed model utilizes hotel reviews as the testing dataset, I recommend that the authors discuss relevant literature on hotel review sentiment analysis and explore the implications or novelty of applying their proposed model to this context.

Reviewer 2 ·

Basic reporting

1. The English language usage throughout the paper is generally professional and clear, but there are areas where the writing could be improved for better clarity and precision. While the technical terms are used appropriately, some sentences are overly complex and could benefit from more concise phrasing. In another hand, some others are too short. A final round of proofreading, ideally by a native English speaker or professional editing service, could help polish the paper, eliminating any remaining awkward phrasing or grammatical errors.

2. The paper includes references to many foundational works in ABSA, WSD, and BERT. However, the literature review could benefit from broader coverage of more recent advancements in the field of sentiment analysis, particularly those involving transformer-based models like BERT. There’s limited discussion of how this work compares to other recent methods that leverage contextualized language models or sentiment analysis improvements in specific domains. The author can expand the literature review to include the latest papers on ABSA that incorporate transformers and other deep learning models (e.g., GPT-based models). Please also provide a more detailed comparison between WSD-BERT and state-of-the-art models beyond BERT, such as models that might use attention mechanisms or hybrid approaches. In advance, emphasize how WSD is particularly valuable over these other models by citing recent work on disambiguation in NLP tasks. The article follows a generally professional structure, with clear sections for introduction, methodology, results, and discussion. However, the presentation of figures and tables could be improved. Some figures lack detailed captions, making it difficult to interpret them without reading the corresponding text in the body of the paper. Additionally, while the data used in the experiments is mentioned, it is unclear whether this raw data has been shared in an accessible format for reproducibility. Please share the raw data by respecting the journal policy.

3. The overall structure/organization of the paper is fine. Nevertheless, some areas could be significantly improved. For example, the author could integrate the first paragraph of the introduction and the "motivation" section to provide a more elaborative and comprehensive outline. Another improvement also can be made in the Literature Review section. While dividing a section into sub-sections can be beneficial to the clarity of paper structure, too many sub-sections with relatively short outlines on each section could also lead to less efficient structure. The author can restructure the literature review to provide a more fluid story of the relevant literature before addressing the current drawbacks of the findings of that literature.

4. In my opinion, the contribution of the paper (lines 116 to 128) should be related to what are the drawbacks identified by the author in recent literature (lines 206 to 212), and the contribution should be placed after the drawbacks identification. However, I found those two is less related, what do you think?

5. Although the experimental results are comprehensive, the choice of evaluation metrics should be justified more explicitly. If any domain-specific benchmarks exist for ABSA, they should be considered, or at least acknowledged if they are not applicable.

6. I am not sure what the mean by term "Ablation Study" is used in this paper, as far as I know, that term is not common. Ain't the content within the "Ablation Study" part of the discussion? If so, then the authors are suggested to integrate them into discussion sections.

7. The paper is largely self-contained, but there are areas where the connection between the results and the original research hypothesis could be strengthened. While the authors propose a novel combination of WSD and BERT, the results section could better highlight whether this combination successfully addresses the stated research objectives. There are also some instances where the discussion is more descriptive than analytical, limiting the paper’s overall impact. Restate the research hypothesis in both the introduction and the results section to ensure the reader is reminded of what the study aims to achieve. The author could also strengthen the discussion of the results by explicitly linking findings back to the research question. For example, explain why certain aspects of WSD improved or did not improve ABSA performance in specific contexts. In advance, please include a more critical analysis of the limitations of the method. What challenges remain unsolved? How can future research build on this work?

Experimental design

1. The paper aligns well with the aims and scope of PeerJ Computer Science, which focuses on the computational sciences, including machine learning, natural language processing (NLP), and applied artificial intelligence. Integrating Word Sense Disambiguation (WSD) with BERT to enhance Aspect-Based Sentiment Analysis (ABSA) addresses a novel problem in computational linguistics, specifically in sentiment analysis. The paper's originality is commendable, but it would be beneficial to highlight more explicitly how this work contributes to the field of NLP, particularly concerning current trends in transformer-based models.

2. The research question is clearly stated, focusing on improving the performance of ABSA by integrating WSD with BERT. However, the articulation of the knowledge gap could be more explicit. While the paper suggests that WSD can resolve ambiguities that affect BERT's contextual understanding, it does not provide a deep discussion of why previous methods fall short and how this particular integration fills a gap in the field. To clarify the contribution, please sharpen the definition of the research question by explicitly identifying the specific limitations of BERT in ABSA, especially in terms of handling ambiguous words or contexts.

3. The investigation appears to be technically sound, with a robust experimental design that includes comparisons between the proposed method and baseline models. However, there is limited information on how the datasets used were curated, whether they are ethically sourced, or if there are any ethical considerations related to data privacy, bias, or model interpretability. Please put an explanation, that addresses the ethical considerations of the research, particularly about the datasets used. For instance, were they publicly available, and were proper licensing and consent obtained where necessary?

4. I am a little confused about the sections "Model Explanation" and "Experiment". The "Model Explanation" section explains the proposed model and how that model was constructed. This is part of the experiment, so why it should be in separate sections? Ain't it be better if those two sections were integrated as the "Methodology" section? The step-by-step figure would be useful to give a clear view of this research. Also, I found some explanations about the concept of evaluation method in the "Results and Discussion" section which are redundant to what is already explained in the "Model Explanation section".

5. Again, the "Model Explanation" and "Experiment" could be merged to provide a more concise structure, and the explanation of each technical approach used is not necessarily that long. In advance, In the methodology section, provide a more granular breakdown of how the integration between WSD and BERT was implemented to ensure clarity for those looking to replicate the process.

6. Some sections of the methodology, particularly the technical steps involved in combining WSD and BERT, could benefit from more granular detail. Moreover, there is little mention of specific hyperparameters, model training processes, or computational resources, all of which are essential for replication. Provide more explicit step-by-step details on how WSD was combined with BERT in the experimental setup.

Validity of the findings

1. The authors predominantly rely on tables of evaluation metrics without offering qualitative examples that demonstrate how their model outperforms existing approaches in handling ambiguous terms. While accuracy scores and other metrics are helpful, they do not provide insight into why or how the model performs better in specific cases. Sentiment analysis models, especially those using WSD, should show examples where they correctly disambiguate a word in different contexts (e.g., "bank" meaning financial institution vs. "bank" meaning a riverbank). The paper does not present any such illustrative examples, which is essential to substantiate their claim that WSD helps BERT better handle ambiguity.

2. Related to previous point 1, The claim that WSD improves BERT’s ability to resolve ambiguities remains largely unproven without concrete examples. For instance, showing an example sentence that BERT alone misinterprets, but the WSD-BERT model correctly disambiguates, would significantly strengthen the claim. The result and discussion section would benefit from showing comparisons between predictions made by BERT alone and the WSD-enhanced model on specific ambiguous words or phrases. This would highlight the contribution of WSD to the overall performance improvement, providing more transparency.

3. The discussion of the results is limited to the interpretation of the metrics (e.g., increased accuracy), but it lacks a critical reflection on why the WSD-BERT model performs better or in which specific scenarios the performance gains are most notable. For a more comprehensive discussion, the authors should explore the limitations of the proposed approach. For example, are there any cases where WSD does not help, or where it even decreases performance? Addressing these points would give a more balanced and insightful discussion of the results.

4. In more general discussion, The paper presents a novel combination of Word Sense Disambiguation (WSD) and BERT for Aspect-Based Sentiment Analysis (ABSA), which holds the potential to advance research in sentiment analysis. However, the novelty and broader impact of the work on the field are not explicitly discussed. The paper does not delve deeply into how the proposed method can inspire further research or how its replication can benefit the literature. Within the discussion, based on the experimental results, the authors are suggested to explicitly assess the potential impact of the study. For instance, how does integrating WSD with BERT improve ABSA in ways that could influence future work in NLP, sentiment analysis, or even other domains of AI? Clarifying this would help readers understand the broader implications of the research.

5. While the paper discusses the datasets used, it does not indicate whether the underlying data, models, and code have been provided for public access. There is a lack of explicit mention of how robust and controlled the data collection process was, or whether potential biases were addressed. Additionally, the statistical analysis performed on the results, while presented, does not appear to fully explore the statistical soundness or reliability of the findings. Hence, the authors could strengthen the discussion around the statistical robustness of the results. For instance, include statistical tests or confidence intervals that support the observed performance improvements. This would give readers a better understanding of the reliability of the results.

6. The paper provides a conclusion that summarizes the research, but there is room for improvement in how tightly it connects the findings to the original research question. The current conclusion does not delve deeply enough into the implications of the results or how they directly answer the research hypothesis. Furthermore, the conclusion sometimes strays into speculation rather than strictly adhering to the results presented.

Additional comments

The paper investigates Aspect-Based Sentiment Analysis (ABSA) and explores the integration of Word Sense Disambiguation (WSD) with BERT, a transformer-based language model. This research is quite timely, given the advancements in natural language processing and the growing need for more accurate sentiment analysis in areas like customer feedback, product reviews, and social media analysis. The paper attempts to address challenges in sentiment analysis by leveraging BERT's contextual understanding alongside WSD for enhanced aspect detection.

Reviewer 3 ·

Basic reporting

- The paper is well-written, and the English is clear and professional. The terminology and technical language are appropriately used for the field of Natural Language Processing (NLP) and Aspect-Based Sentiment Analysis (ABSA).

- The paper offers a solid background on sentiment analysis and ABSA, providing sufficient context for readers familiar with NLP. However, the literature review could benefit from additional references to more recent works, particularly in the areas of Word Sense Disambiguation (WSD) and its integration with advanced neural architectures such as BERT and Graph Convolutional Networks (GCNs). Expanding on prior research would better position the proposed method within the existing body of work.

- The paper follows a professional and logical structure, with clear sections delineating the introduction, methodology, experiments, and conclusion. The use of figures and tables to present experimental results is appropriate, though the clarity of some visuals (e.g., diagrams of the model architecture) could be improved for easier interpretation. Raw data from the RABSA dataset is mentioned but not directly shared; ensuring that the dataset or detailed access instructions are available would enhance reproducibility.

- The research is self-contained, with the results effectively addressing the proposed hypothesis. The integration of WSD into ABSA is novel and meaningful, with the results showing the value added by this technique. The ablation study further supports the relevance of the WSD module.

- The paper provides clear definitions of key terms such as Word Sense Disambiguation (WSD), Bidirectional Encoder Representations from Transformers (BERT), and Graph Convolutional Networks (GCNs). While formal theorems are not applicable in this context, the results are grounded in empirical experiments, which are well-documented. The method section could benefit from more precise definitions of certain terms (e.g., “aspect” and “similarity measures”) for clarity.

Experimental design

- This research fits well within the aims and scope of journals focused on NLP, machine learning, and applied AI in real-world contexts, such as hospitality and consumer reviews. The paper contributes original work by introducing a novel method combining WSD, BERT, and GCN for ABSA, filling a gap in the sentiment analysis literature by improving the understanding of word sense and aspect relations.

- The research question is well defined: how can the integration of WSD with neural network architectures improve aspect-based sentiment analysis? The paper clearly outlines how the proposed method addresses an existing gap by resolving word ambiguities in sentiment analysis. The novelty of combining WSD with BERT and GCN is justified as filling this gap.

- The research demonstrates a high technical standard, employing a combination of state-of-the-art NLP techniques, including WSD, BERT, and GCN. The experiments are robust, and the results are thoroughly analyzed. However, the ethical considerations, such as the use of human-generated review data, are not explicitly discussed. It is recommended to briefly mention any ethical guidelines followed, especially regarding data privacy.

- The methodology is generally well-described, allowing for replication by other researchers. The integration of WSD with BERT and GCN is detailed, and the ablation study strengthens the paper’s claims. Some aspects, such as parameter settings and training details, could benefit from further elaboration to ensure full replicability.

Validity of the findings

- The paper refrains from over-assessing its impact and novelty, which is appropriate. Meaningful replication is encouraged, with the authors stating that the method can be applied to other sentiment analysis tasks. The rationale for replication is sound, given the method's potential application in various domains beyond hospitality.

- The paper provides experimental results that are statistically sound, with appropriate baselines for comparison. The robustness of the results is demonstrated through the ablation study, which confirms the contribution of the WSD module. However, the raw data (RABSA dataset) should be made more accessible to fully support the reproducibility of the study.

- The conclusions are clearly stated and align well with the research question and results. The paper emphasizes the importance of WSD in improving ABSA performance and highlights its practical implications for the hospitality industry. The conclusions are appropriately limited to the findings of the experiments and do not overstate the results.

---

## Round 0.2 · Minor Revisions

Dear Authors,

Thank you for submitting your revised article. Feedback from the reviewers is now available. We strongly recommend that you address the issues raised by Reviewer 1 about reflecting the explanations in the actual manuscript and resubmit your paper after making the necessary changes.

Best wishes,

Reviewer 1 ·

Basic reporting

see additional comments

Experimental design

see additional comments

Validity of the findings

see additional comments

Additional comments

I appreciate the authors' response to my comments. Most of my comments were addressed and explained by the author in the rebuttal letter and the revised manuscript. First, the author better positioned the paper and highlighted the contribution around the WSD. Second, regarding my comment on data collection, the authors provided a detailed response that is satisfactory; however, I did not see these explanations reflected in the manuscript. It would be better if the author made this clear and reflected these changes before final acceptance. Third, the paper's presentation and organization have been improved. Fourth, the rationale for not reporting the category results is acceptable, but it would be better to reflect this in the final manuscript.

In short, most of the major concerns have been properly addressed, and there are a few minor issues that I believe could be addressed in a minor revision cycle.

Reviewer 2 ·

Basic reporting

The authors have addressed all the revision issues well

Experimental design

The authors have addressed all the revision issues well

Validity of the findings

The authors have addressed all the revision issues well

Additional comments

The authors have addressed all the revision issues well

Reviewer 3 ·

Basic reporting

No comments.

Experimental design

No comments.

Validity of the findings

No comments.

Additional comments

No comments.

---

## Round 0.3 · accepted · Accept

Dear Authors

Thank you for addressing the reviewers' comments. The manuscript now seems ready for publication.

Best wishes,

Reviewer 1 ·

Basic reporting

See my comment in the section "additional comments"

Experimental design

See my comment in the section "additional comments"

Validity of the findings

See my comment in the section "additional comments"

Additional comments

Thank you again to the author for the responses. The final two comments from the second-round review have been addressed and are satisfactory.